# A Case Study of a Natural Experiment Bridging the ‘Research into Policy’ and ‘Evidence-Based Policy’ Gap for Active-Living Science

**DOI:** 10.3390/ijerph16142448

**Published:** 2019-07-10

**Authors:** Paula Hooper, Sarah Foster, Billie Giles-Corti

**Affiliations:** 1Australian Urban Design Research Centre, School of Design, The University of Western Australia, Perth 6009, Australia; 2Centre for Urban Research, RMIT University, Melbourne 3000, Australia

**Keywords:** natural experiment, built environment, urban design, policy evaluation, active living, liveability, Australia

## Abstract

The translation of research into tangible health benefits via changes to urban planning policy and practice is a key intended outcome of academic active-living research endeavours. Conversely, policy-makers and planners identify the need for policy-specific evidence to ensure policy decisions and practices are informed and validated by rigorously established evidence. In practice, however, these two aspirations rarely meet and a research-translation gap remains. The RESIDE project is a unique longitudinal natural experiment designed to evaluate the health impacts of the ‘Liveable Neighbourhoods’ planning policy, which was introduced by the Western Australian Government to create more walkable suburbs. This commentary provides an overview and discussion of the policy-specific study methodologies undertaken to quantitatively assess the implementation of the policy and assess its active living and health impacts. It outlines the key research-translation successes and impact of the findings on the Liveable Neighbourhoods policy and discusses lessons learnt from the RESIDE project to inform future natural experiments of policy evaluation.

## 1. Introduction

Government policy and planning initiatives determine the way cities, towns and neighbourhoods are developed and configured. They also play a vital role in creating and shaping the environments that support or undermine residents’ health [1] and their ability to be safely and conveniently physically active [2,3]. Creating healthy, active communities is recognized as a global priority from both environmental sustainability and health perspectives [2,4]. However, achieving this laudable goal is not without challenges. It requires the involvement (and commitment) of multiple sectors beyond health, including urban planning and design, property development, construction, and finance, each with competing priorities and pressures.

Over the last 15 years, a comprehensive body of research has documented the impact of the design of the built environment on residents’ active-living behaviours [5,6,7,8]. The translation of active-living research into tangible health benefits via changes to urban planning policy and practice is a key intended outcome of these academic endeavours [9]. At the same time, enlightened policy-makers and planners regularly affirm the need for “evidence-based policy and practice” to ensure their policy decisions and practices are informed and validated by rigorously established evidence. In practice, these two aspirations are rarely met and a research-translation gap remains between the ambitions of public health and urban planning on the one hand and the on-ground delivery of healthy, active communities on the other. With an increasing awareness and acceptance of the need for health-enhancing planning interventions, it is essential to understand why this research-translation gap remains. One explanation relates to the type of health-related evidence that is needed by planners and (planning) policy-makers in order for it to be utilized and applied [6]. This raises the question, is all evidence equal? Allender and colleagues [10] have argued that ’evidence-based‘ public health recommendations to planners and policy-makers are usually made without any obvious links to existing policies or legislation. Moreover, public health evidence rarely (if ever) provides quantifiable, evidence-based information about the potential health impacts of urban planning policies and decisions [10,11]. This has led to calls for public health research to be better aligned with current and future policy environments [6,12], and for science to more effectively guide city planning policy and practice [13]. However, if science is to inform policy, it must match the policies that it aims to influence [13]. Studies that develop policy surveillance measures and investigate and evaluate planning policy processes and implementation are well placed to bridge the translation gap [13,14]. Such research can provide a better, more nuanced understanding of how policies are implemented, how much of the policy has been delivered (i.e., the ‘dose’ of the policy intervention) and allow more accurate quantification of its impacts on active-living behaviours [14]. In particular, case studies and evaluations of urban planning policies, undertaken in partnership with planning professionals, are needed to identify the policies (or parts thereof) that produce desirable health-related outcomes [13,14,15].

To date, the public health evidence has mainly been cross-sectional in nature and the field has largely focused on the need for longitudinal natural experiment studies to evaluate policy implementation and as a way of understanding the impact of population-level policies on health outcomes [16]. A growing number of case studies and research papers now exist which compare new areas developed under different design principles or alternate planning movements or theories such as New Urbanism or Smart Growth and compare the health outcomes of residents [17,18,19]. These studies have typically measured and characterized the built environment with regards to how it relates to New Urbanist principles or planning policies [19,20]. However, to our knowledge, no studies have explicitly assessed or quantified the implementation of specific planning policies or design codes and empirically evaluated their impact on health-promoting behaviours and positive wellbeing outcomes. This is despite calls for studies of this type [6].

In addition to prospective study designs, greater emphasis is needed to identify which aspects of the policy produce desirable health-related outcomes [6,10,13,14,15,21]. This requires additional emphasis on process evaluation. Policy evaluation enables detailed assessment and quantification of which components of a policy were implemented as intended (in order to assess the ‘dose’ of the policy implementation), which components of the policy are the most influential active living ingredients, and which have no observable impacts. Without process evaluation, it is impossible to know whether positive effects (e.g., desirable active living or health behaviours) are the result of the policy intervention itself, and it is also impossible to establish whether a lack of observed outcomes is due to policy failure or inadequate policy implementation [22].

Few public health studies have measured policy implementation using policy-specific measures monitored over time, and assessed their impact on health-supportive behaviours [21]. Planning academics have also lamented the dearth of studies that quantitatively assess the implementation of policy to determine to what degree urban development policies and guidelines have been implemented as intended, how this relates to the intended planning goals, and to assess the on-ground outcomes [23,24]. Indeed, much of the literature about the implementation of planning theory concentrates on examining planning documents, processes, or decisions [18], not what is actually implemented in reality in communities and neighbourhoods.

State and local governments in jurisdictions around the world, including North America and Australia, are seeking to implement urban planning policies designed to shift growth away from low density, automobile-oriented development [6]. While evidence from across the globe in different settings is an important input which informs local decision-making, policy-makers often prioritize local evidence that directly relates to the planning and policy context [9]. Different jurisdictions in different contexts implement different policies and, as a result, produce different outcomes [25]. Hence, creating context-specific, local, policy-relevant evidence on the implementation of local policies is both important and timely.

## 2. RESIDE—A Natural Experiment of a New Urban Planning Policy

In 1998, when the State Government introduced the Liveable Neighbourhoods Community Design Guidelines policy (LN), a unique opportunity arose to collaborate with policy-makers and practitioners to evaluate the health impacts of an urban policy reform in situ through a longitudinal natural experiment study design. [26]. Liveable Neighbuorhoods was a response to conventional planning policies and practices implemented through the 1970s and the 1990s that had facilitated suburban sprawl and motor vehicle dependency across Perth, the capital city of Western Australia. The LN policy embraced emerging New Urbanism planning concepts [27] that provided an alternative approach to suburban neighbourhood design. LN has promoted a structure of walkable neighbourhoods where community facilities and services are (ideally) accessed by walking, cycling, and public transport through an efficient, interconnected movement network.

In 2003, with the support of key local policy-makers and advocates, the RESIDential Environments project (RESIDE) began with the aim of assessing the impact of LN on the desired policy outcomes including walking, cycling and public transport use, sense of community, safety from crime, and mental health.

All “liveable” and “conventional” developments that were under construction across the Perth metropolitan region with land sold for housing during the RESIDE recruitment period (in 2003) were included in the study. The Western Australian Department of Planning, Lands and Heritage (formerly the Department of Planning) categorized new development applications it received as either ‘liveable’ (i.e., aspiring to meet many of the LN requirements), ‘hybrid’ (i.e., meeting some but not all of the LN requirements), or ‘conventional’ (approved under the old policy). A total of 74 new developments (19 LN, 11 hybrid, and 44 conventional) [28] were selected for inclusion in the longitudinal study based on their stage of development, size, and location (e.g., distance to the ocean). The majority were being constructed on greenfield sites (i.e., previously unused or undeveloped land areas that had been rezoned, typically from urban deferred or rural to urban land uses and projects). Others were being constructed in brownfield areas (existing urban zones being redeveloped, sometimes following rezoning from industrial or other non-residential use).

The Water Corporation was approached to assist with the study. They invited all households that purchased house and land packages in the 74 developments (*n* = 10,193) to participate in the study [28]. To be eligible, customers had to be ≥ 18 years of age; proficient in English; building a home in the selected development and planning to move into that home by December 2005; and they had to indicate that they were willing to complete three surveys and wear a pedometer for one week on three separate occasions over a five-year period. Only one eligible person was selected at random from each household [28]. The University of Western Australia’s Human Research Ethics Committee provided ethics approval (#RA/4/1/479). Participants completed a postal survey on four occasions: Time 1 (T1), at baseline in 2003–2005 during construction of their new home and before relocation (*n* = 1813; 33.4% response rate); Time 2 (T2) in 2004–2006, approximately one year after relocation to their new home (*n* = 1465); Time 3 (T3) in 2006–2008, approximately three years after relocating (*n* = 1229); and Time 4 (T4) in 2011–2012, about six to nine years after relocating (*n* = 565) [29].

The mean age of the study population at baseline (*n* = 1813) was 40 years. Sixty percent were female, 82% were married or living with a partner, and 49% had children living at home. Just under one quarter of the sample (23%) had a bachelor’s degree or higher, 43% had professional or managerial-administrator occupations, and 25% lived in households that earned AUD$90,000 or more per year. A large proportion of participants (43%) worked 39–59 hours per week and almost all (98%) had access to a motor vehicle [28,29].

This commentary provides a case study overview and discussion of the key lessons from the RESIDE project. It focuses on the policy-specific methodology and analyses that ensured RESIDE findings resonated with the local planning industry, and our research-translation efforts and their subsequent impacts on the LN policy. In discussing RESIDE we will also reflect on the three domains of evidence-based policy identified by Brownson and colleagues [15] and five key strategies identified as being essential to close the active-living research-translation gap [9]. The alignment between these domains and translation strategies is outlined in Table 1.

## 3. Understanding the ‘Policy World’ for Liveable Neighbourhoods

In earlier work [9], we highlighted the critical need for active-living researchers to understand the ‘policy world’ in which decisions are made, and which active-living researchers are trying to influence. Similarly, Gagnon and Bellefleur [30] argue that public health researchers need to be familiar with the science of public policy-making in order “to better understand potential intervention contexts”. Research-translation opportunities, therefore, need to be understood in the context of politics and public policymaking processes. Policy frameworks developed in the public policy and political science fields are used in a range of disciplines to analyze agenda-setting and policy-making. The Multiple Streams Framework originally developed by Kingdon [31] is an agenda-setting theory focused on how major policy change comes about. It posits that three elements contribute to policy change: a problem stream consisting of the issues that policy-makers and citizens want addressed; a politics stream comprising the factors that affect policy-makers’ willingness to make a decision, including pressure group campaigns and political ideology; and a policy stream made up of ideas for feasible policy solutions [31]. According to this theory, policy change is possible when these streams align at critical moments in time, opening a ‘window of opportunity’ with the help of one or more policy entrepreneurs (individuals or organizations) who act as power brokers and manipulators of preferences.

The introduction of the LN policy was the result of a ‘window of opportunity’ where, in accordance with Kingdon’s three streams theory of policy development, there was convergence between the problem, policy, and politics [31]. First, the problem of unsustainable suburban development and sprawl (the problem stream) was recognized and perceived as an issue of pressing importance for the future sustainability of Perth by policy-and decision-makers. Second, a proposal for an alternative policy solution promoting more sustainable design principles to those currently in operation was introduced via the global advocacy efforts of the Congress of New Urbanism (the policy stream). Finally, there was a supportive political climate and a willingness to trial a new approach (the politics stream) from the State Government, the Western Australian Planning Commission and the Department of Planning, plus growing awareness for the need of a different planning approach among the more enlightened property developers, through the advocacy and educational efforts of Council of New Urbanism [27]. The convergence of the three streams opened a policy window [31], allowing the trial of a new policy—the Liveable Neighbourhoods design guide.

The first edition of the new LN policy instrument was introduced in 1998 as a trial voluntary design code to be implemented by developers. The policy underwent extensive public and industry consultation before being adopted for full implementation in 2002 (as Edition 2) by the Western Australian Planning Commission as its preferred policy for assessing and approving all new greenfield and infill development applications in Western Australia.

In 2013, a decade after the RESIDE team first surveyed its cohort of participants and 15 years after the introduction of the LN policy, a comprehensive review of the LN policy was announced by the West Australian Department of Planning. This presented the opening of the policy window for the results from RESIDE to inform and influence the LN policy.

## 4. Establish Links and Joint Research Agendas with Policymakers and Practitioners

A partnership with the Department of Planning was formed in the planning stages of RESIDE, which was instrumental in galvanizing the Department’s support for the project. Departmental officials were involved in selecting the developments for inclusion in the study, and later, a multi-sector advisory board was established to oversee and advise on the direction of RESIDE as it progressed. Throughout the study, Department of Planning staff received regular project updates. As a result, they were privy to results prior to publication and were able to provide feedback to the RESIDE study team. This process helped researchers to articulate the planning and policy implications of their research, and embed these in the ‘so what’ messaging and communication of the findings. Further, key supporters of the project from the Department provided opportunities for RESIDE researchers to present updates and findings to Department staff. This was essential in generating ongoing exposure and keeping the project on the Department’s agenda.

Whilst the LN was highly regarded internationally as an effective local interpretation of New Urbanism, and despite the (then) Minister for Planning stating, in his foreword of the 2000 (second) edition of the guidelines, that *“the ability to measure their on-ground performance will further refine the policy”* [26], by 2003 there had been no formal evaluation of the policy. Therefore, in addition to assessing the health and wellbeing outcomes of the LN policy, a key interest of the policy partners within the Department was the assessment of which aspects of the policy were (or were not) being implemented and were (or were not) most effective. Upon the commencement of the review of the LN policy in 2013, and as a consequence of the ongoing collaboration and communication with the Department of Planning, there was, therefore, considerable interest in identifying which, if any, of the multitude of design features addressed in the policy were the ‘key performance indicators or “non-negotiable” requirements for enhancing health’.

## 5. Quantify Policy Implementation and Delivery

In order to accurately assess the impacts of the LN policy on the health behaviours of residents and evaluate whether the policy was achieving its intended outcomes, it was essential to determine the degree to which the on-ground implementation had occurred (i.e., the ‘dose’ of the policy intervention that had been delivered). Hence, a novel aspect of the RESIDE project was the inclusion of a process evaluation to measure and quantify the levels of on-ground delivery of the policy in a subset of 36 of the 74 housing developments. The 19 LN developments were matched with 17 of the 44 conventionally designed developments by their stage of development (i.e., the proportion of the gross development area that had been constructed), size, and location (i.e., distance from the ocean). The varied sizes provided an opportunity to investigate how the LN policy was being applied at different scales of development and at which scale (or scales) the policy produced the greatest impacts.

Although RESIDE was a longitudinal study, the process evaluation of the policy implementation was cross-sectional in design. The majority of the housing developments selected for inclusion in the RESIDE project were being built on greenfield sites (i.e., previously undeveloped land). Because of the lack of any existing infrastructure on these sites and the timelines required for the construction of the developments from scratch, the timing of the process evaluation was chosen to coincide with the third time point of RESIDE data collection—i.e., five to six years post the commencement of the RESIDE study and approval of the housing developments. This allowed for the greatest amount of construction to have occurred within the study housing developments.

The process evaluation was designed to understand how much of the policy and what specific design requirements were (or were not) being adopted and implemented, and whether any dissappointing observed health and wellbeing outcomes were due to shortcomings of the policy principles, or failure to implement the policy [32,33]. The process evaluation used spatial measures tailored to quantify the urban design features required by the policy [32]. A total of 43 requirements were measured across four policy elements:(1)Community design *n* = 13: These requirements determined the provision, location, and configuration of neighbourhood centres to create a hub of diverse destinations that attract people to a variety of activities. Objective measures of design features included land-use mix, number of destinations, design and configuration of activity centres, and presence of schools;(2)Movement network *n* = 15: These requirements aimed to produce a highly interconnected street system aimed at reducing travel distances to local centres, schools, public transport links and other destinations, and adequate infrastructure for pedestrians and cyclists. Objective measures of design features included street connectivity, cul-de-sac lengths and block sizes, footpath networks and public transport;(3)Lot layout *n* = 7: These requirements focused on higher residential densities to create more compact urban development and encourage the provision of a mixture of residential lot sizes to facilitate housing variety, choice and affordability, and to cater for increasingly diverse household types. Objective measures of design features included residential density, average area, and mix of residential lot sizes;(4)Public parkland *n* = 8: These requirements aimed for a minimum contribution of 10% of the gross subdivisible land area to be provided as public parkland (Western Australian Planning Commission 2000) and specify different park types based on size and catchment areas to provide for a range of uses and activities. Objective measures of design features included the number and area of public parkland, distance to parks, and the assets and amenities within the parkland.

Policy compliance was defined as the degree to which the LN standards or requirements were reflected in the ‘on-ground’ construction of the developments. A simple scoring system was developed to quantify the extent to which the 43 measurable requirements had been implemented as intended by the LN [32]. The level of compliance for each element (defined as the degree to which the developments met the LN standards within that element) and overall LN compliance was calculated as the percentage of the maximum policy implementation score attainable.

The findings revealed that none of the developments had implemented the full suite of requirements as intended by the policy [32] and indeed, across all new developments, overall the LN policy was only half implemented, with overall compliance averaging just 46% (range: 30–60%) across the 36 developments [32]. Percentage compliance scores for each of the four elements were also well below full implementation: community design 27% (0–67%); movement network 48% (37–59%); lot layout 52% (19–88%); and public parkland 48% (30–60%) [32].

Figure 1 identifies the policy targets for seven design requirements from the LN policy that were found to be supportive of health behaviours [32,33,34,35,36]. It also plots the measured levels of on-ground compliance in each of the 36 RESIDE housing developments that were included in the process evaluation [32]. The results were presented in this way to the Department of Planning to clearly represent the levels of implementation being achieved versus the policy target and aspiration, and how this differed between design requirements.

## 6. Quantify Policy Impacts on Health-Related Outcomes

Despite incomplete implementation of the policy, analyses were also undertaken to determine if greater on-ground implementation of the policy was associated with positive health and wellbeing outcomes. The results revealed a strong dose–response relationship between policy compliance and four health-related outcomes, suggesting that new communities built in accordance with the LN policy principles and design features have the potential to promote the health and wellbeing of residents by creating neighbourhoods that encourage transport and recreational walking and have a stronger sense of community where residents feel safer [32,34,35,36]. For every 10% increase in levels of overall policy compliance, the odds of RESIDE participants walking for transport in the neighbourhood increased by 53% [32]; the odds of having a higher sense of community increased by 21% [35]; low psychological distress (i.e., better mental health) increased by 14% [35]; and the odds of being a victim of crime decreased by 40% [34]. Furthermore, a series of analyses unpacked the policy further to identify which of the specific design requirements from each of the policy elements were most strongly associated with walking [36], sense of community, mental health [35] and reports of safety or being a victim of crime [34]. Additional analyses also investigated which of the specific design features from the four policy elements showed the strongest associations with these four outcomes [32,34,35,36]. Other RESIDE analyses have also shown the design of the neighbourhoods to be positively associated with public transport use [37] and cycling [38].

## 7. Understand the Barriers and Facilitators to Policy Implementation

The results of the process evaluation revealed incomplete levels of implementation and compliance with the LN policy in that the totality of on-ground built form outcomes intended by the LN was at the time, not evident. This raised questions as to why incomplete implementation had occurred. Because the planning principles underpinning the LN differed from the conventional planning policies and development and engineering practices of the time, problems were experienced when implementing the LN.

Working with the project advisory board and key personnel from the Department of Planning, considerable efforts were made to identify and understand the barriers or factors that limited or acted as obstacles to the implementation of the LN. This was also an essential step in helping to determine whether principle or practice gaps were affecting its success. Conversely, it also helped to identify the factors or approaches that enhanced the likelihood of policy adoption and implementation.

For example, the construction of housing developments is generally sequenced, and the order in which land and infrastructure are developed appears to be dictated by several factors, including a balance between marketing or sales purposes and economic considerations. Developers appear to regard public open space as an important aesthetic feature that is instrumental to land sales and as such is typically installed early [39,40]. In contrast, community infrastructure such as neighbourhood centers, health services, schools, and public transport are often delayed until there is a sufficient critical mass of residents to warrant these services being provided [29,41,42]. The longitudinal component of the study allowed these changes in the new developments to be identified, tracked, and explained over time.

The LN policy document is also complex. It contained 128 different requirements and significant amounts of duplication within and across the four elements, which proved to be daunting and difficult for developers to understand and apply [43].

Furthermore, development proposals submitted for assessment and approval under the LN policy required additional compulsory information compared with the conventional policies. This was a significant disincentive for developers and resulted in their reluctance to submit proposals using the new LN policy, which also created difficulties for officials assessing and approving the applications [43].

Many of the LN design requirements directly contradicted existing conventional policies and engineering standards. Whilst LN was meant to prevail in instances of conflict, the voluntary nature of the guide meant it had no legal standing or precedence over the existing Local Government Authority planning schemes. It was, therefore, vulnerable to negotiation and to the compromising of LN standards. As a result, many of the LN features approved for development may not have been implemented on the ground. The inconsistencies between the state-sanctioned LN and LGA planning schemes were identified as a major barrier to developers adopting and implementing the LN policy (UDIA 2005; Jones 2010; STAC 2012).

Finally, given the time taken to construct the new developments, the six years between the commencement of the RESIDE project (in 2003) to the time of evaluation (in 2009) was a relatively short time period. This is likely to have contributed to many of the developments being incomplete at the time of evaluation. This was an important consideration not to be overlooked when designing future natural experiment studies of policy implementation.

The findings indicated that the policy was worthy of wider dissemination, but a greater emphasis on policy implementation was needed. The identified policy implementation gap highlighted the importance of process evaluation and the need for longitudinal study design. It also highlighted the value of undertaking research in partnership with policy-makers within local contexts [9,10,11].

## 8. Highlight Specific Policy Implications

When undertaking case studies or natural experiment evaluations of planning policies and practices in partnership with policy-makers and practitioners, outputs that are directly relevant to the policy and its implementation are needed to help partners gauge the health impact of their current policies and practices. When presenting updates and results to our industry advisory board, we were consistently challenged to reflect on the findings and identify the policy significance in terms of being able to communicate ‘what bit of what policy would we change and what should it be changed to?’

RESIDE’s process evaluation study provided policy-specific empirical evidence that showed that when implemented as intended, the LN policy could positively impact a range of health and wellbeing outcomes and produce outcomes that were aligned with the policy’s overall objectives. Through the quantification of policy implementation with tailored spatial measures and the longitudinal tracking of the built environments over time, the RESIDE process evaluation project highlighted important gaps in the implementation of LN design features. Without this knowledge, it was impossible to judge whether the observed associations with health outcomes (or lack of) were due to the ineffectiveness of the policy principles or a failure to implement the policy on the ground.

This data resonated with the Department of Planning, which was keenly interested in the policy-specific measures of implementation as they both illustrated and confirmed how well the policy was being implemented and the potential promise of the policy principles. For the first time, the process evaluation armed the Department of Planning with the objective evidence they needed to assist in reviewing a planning policy and its processes. Importantly, it enabled them to gauge current levels of LN implementation and identify what aspects of the policy were (or were not) being delivered on the ground.

In direct response to their interests and jointly established research questions, we were able to identify the specific design requirements shown to be important for health and wellbeing outcomes, derived from analyses examining individual policy requirements [32,33,34,35,36]. The empirically based submissions were also important in terms of preventing policy regression (i.e., health-promoting requirements were not removed from the policy). Specifically, policy advocates needed a highly regarded, policy specific, and well-understood evidence-base to help preserve the design requirements that had been shown to consistently and positively influence health-enabling or health-promoting behaviours. This was an important outcome of the RESIDE project and a direct result of the policy-specific nature of the measurement and analyses undertaken.

## 9. Undertaking Policy-Relevant Research and Natural Experiments—Lessons Learned

The long-term partnership between researchers and policy-makers from the Department of Planning was paramount to the success of the RESIDE research-translation. Throughout the study, Department of Planning staff received regular project updates, ensuring they were privy to results prior to publication, and provided feedback to the RESIDE study team. This process helped researchers to understand and articulate the planning and policy implications of their research and embed these in the ‘so what’ messaging and communication of the research findings.

Another crucial success factor was understanding the planning and development processes. Concerted efforts were made to work with our industry partners to understand the development and construction processes of the local planning system. Indeed, the relevance of research findings and their applicability to a local context have been identified as important enablers of research uptake among policy-makers (Oliver et al. 2014). The guidance provided by the partners was essential in helping the research team to understand (1) the different stages of the ‘policy pipeline’—that is, the different stages of approval a development application goes through; (2) the relevant authorities responsible for the approval of a development application at each of stage of the process (i.e., local government versus state government); and (3) which specific design requirements were assessed and approved at each stage of the process and at different scales of development (e.g., regional masterplan, structure plan, or subdivision). This knowledge was essential in alerting the researchers to the challenges that practitioners face throughout the development application process, whilst simultaneously helping and enabling the research team to credibly frame, contextualise, and communicate the policy-relevance of the findings when presenting to the policy-makers and planning practitioners. Further, it was apparent that different aspects of policy compliance were the responsibility of different authorities, and it was important to understand this to tailor targeted messages to specific groups. Other lessons learned were an understanding of the complexity of the process of developing policy and delivering outcomes on the ground in communities, and a recognition of the number of actors involved in the policy pipeline. With our industry partners, we attempted to conceptualize this process in what we’ve termed ‘the leaky pipe’ of the policy pipeline process (see Figure 2).

The concept of the ‘leaky pipe’ in policy evaluation studies is important, because if policies are to be fully implemented and studies are to be established to identify where leakages occur between a policy and its on-the-ground delivery, it is essential that researchers understand what agencies or authorities are responsible for the approval and enforcement of different design features within the policy. Moreover, understanding the policy pipeline could help with the translation of active-living and public health research findings by going beyond simply identifying that a policy was not working to identifying where in the policy implementation pipeline leakages were occurring. This would enable specificity in the messaging by highlighting the aspects of the policy that are important for positive health impacts. Moreover, if different organizations are responsible for different design features, it may be necessary to communicate and target research findings to the specific authorities and agencies responsible for these design requirements at different stages of the approvals process.

In an ideal world, there should be an increase in policy compliance at each stage of the policy pipeline, as the different authorities uphold and enforce the intent (and implementation) of the policy. But at each stage, there is also the potential for ‘leakage’ (e.g., failure to implement parts of the policy). Hence, research is needed to understand and identify how much and what specific features of the policy are being approved and enforced at each stage of the process, to enable any leakages to be ‘plugged’ or resolved to ensure full policy implementation. Moreover, if the custodians of the policy (in our case, the Department of Planning) wish to close the gap between policy and implementation, they need to exercise greater oversight of development applications as they move through each stage of the pipeline process and identify and stem any leakages.

Our process evaluation provided a benchmark of the on-ground delivery of the LN policy for the Department of Planning; however, they have since embarked on an ongoing (in-house) monitoring program to track LN policy uptake. In partnership with department planners, we identified the design requirements from the LN policy that had been shown (from RESIDE) to be important for the health and wellbeing outcomes that they were directly responsible for enforcing. These were adopted by the Department of Planning in 2016 as ‘performance indicators’ to provide a framework for the assessment of future development applications against the LN policy. This consistent capture of policy-specific data by the Department was also the basis for an ongoing monitoring and evaluation framework for the LN policy, and a cross-check of how well (and how much of) the policy was being adopted and enforced at the last point in the pipeline before construction.

A final lesson learned from RESIDE was the need to benchmark and quantify the levels of compliance against the policy aspirations and targets at different stages of the policy pipeline. This is important to explain findings related to the intended policy outcomes versus those actually being realized. It would also assist in translating research findings, ensuring that different authorities are appropriately targeted to assist in avoiding leakages between policy and delivery on the ground. Future process evaluations of policy implementation could seek to quantify the levels of compliance with the policy at the different stages of the policy pipeline with the aim of identifying where leaks in the system are occurring and to gain an understanding of the barriers to, or facilitators of, compliance and implementation.

## 10. Conclusions

This paper demonstrated how policy-relevant research and natural experiments can be undertaken and disseminated to policy-makers to positively impact policy. The longitudinal natural experiment approach adopted by RESIDE allowed time for sufficient development to unfold. This, coupled with a process evaluation to quantify the dose of the LN policy being evaluated, helped to bridge the gap between active-living research and its application for evidence-based (or informed) planning policy. Evidence-informed planning and better monitoring of urban policy implementation [32] can help assess progress towards maintaining and strengthening the health and liveability of cities. The development and regular measurement of spatial indicators to benchmark and monitor the implementation of policies designed to create healthy, liveable communities is essential to ensure the policies aimed at creating these environments are fulfilled.

## Figures and Tables

**Figure 1 ijerph-16-02448-f001:**
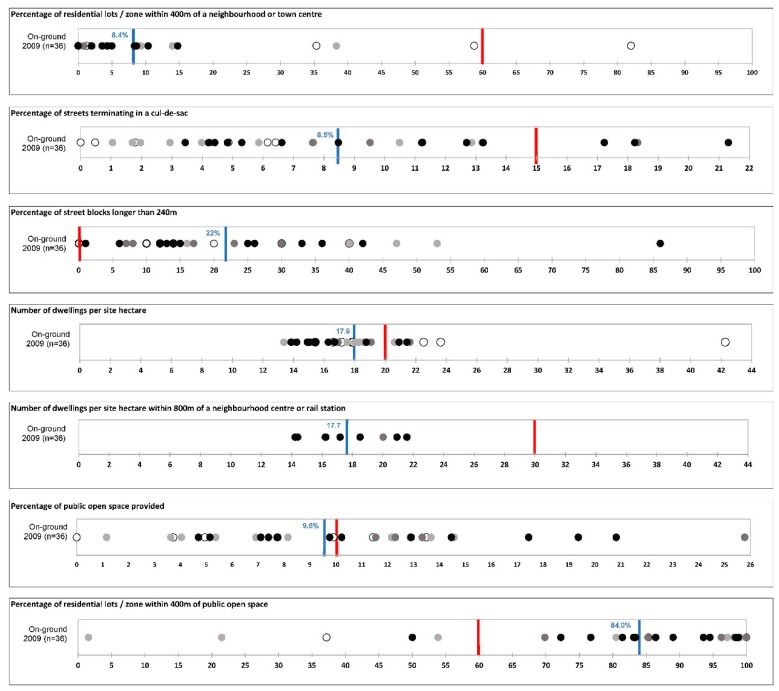
Compliance scores on selected design requirements measured in 36 housing developments on-the-ground in 2009 and approved development applications in 2015. Top = on-ground 2009 (*n* = 36): ○ Small subdivision ≤60 hectare (*n* = 8); ● Large subdivision >60 hectares ≤100 hectares (*n* = 10); ● Structure plan >100 hectares ≤300 hectares (*n* = 5); ● Regional master plan >300 hectares (*n* = 13); Indicates the Liveable Neighbourhoods policy target for the respective design feature; **|** Indicates the average level of compliance across the 36 housing developments.

**Figure 2 ijerph-16-02448-f002:**
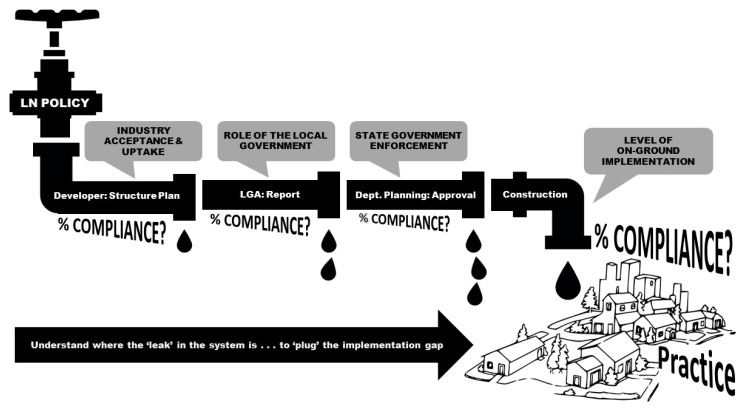
The ‘leaky pipe’ of the policy pipeline process for the Liveable Neighbourhoods Community Design policy in Perth, Western Australia.

**Table 1 ijerph-16-02448-t001:** Criteria for assessing the policy-relevant research-translation efforts of the RESIDE project.

Three Domains of Evidence-Based Policy [15]	Strategies for Closing the Active-Living Research-Translation Gap [9]
▪ Process—to understand approaches to enhance the likelihood of policy adoption.	▪ Understand the ‘policy world’ we want to change. ▪ Establish links and research agendas jointly with policy-makers and practitioners. ▪ Apply policy-relevant study designs (e.g., quantifying policy implementation) that evaluate policy reform.
▪ Content—to identify specific policy elements that are likely to be effective.	▪ Identify reasons for implementation (or non-implementation).
▪ Outcomes—to document the potential impact of policy.	▪ Quantifying policy impacts on health. ▪ Highlight specific policy implications.

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
