# Peer review of "A Case Study of a Natural Experiment Bridging the ‘Research into Policy’ and ‘Evidence-Based Policy’ Gap for Active-Living Science"

_ijerph, 2019, doi:10.3390/ijerph16142448_

Round 1
Reviewer 1 Report
This is a well written manuscript that adds value to translation into policy and practice of active living research. The manuscript reports on the processes undertaken to translate the findings of the RESIDE study into policy and practice. I have only a few very minor edits to address.
- Line 38 - 'met' rather than 'meet'
- There are extra spaces between sentences throughout - please delete these
- Line 90 - 'Australia' not 'Australian'
- Figure 1 and 2 are blurry and unable to be read
- Line 304 - insert 'of' before 'an ongoing'
Author Response
Reviewer 1:
Comments and Suggestions for Authors
· This is a well written manuscript that adds value to translation into policy and practice of active living research. The manuscript reports on the processes undertaken to translate the findings of the RESIDE study into policy and practice.
· I have only a few very minor edits to address.
- Line 38 - 'met' rather than 'meet' – this has been corrected
- There are extra spaces between sentences throughout - please delete these – this has been corrected
- Line 90 - 'Australia' not 'Australian' – this has been corrected
- Figure 1 and 2 are blurry and unable to be read – this has been corrected
- Line 304 - insert 'of' before 'an ongoing' – this has been corrected
Reviewer 2 Report
Comments to the Authors:
Overall Comments:
This paper provides methods, lessons learned and results from a longitudinal natural experiment that rigorously evaluated the implementation process to quantify the dose of the policy and understand the relationship to the intended active living and health outcomes. The insights from this study are applicable and important for bridging the research- policy gap.
There are several grammatical edits, typos, etc. that need to be fixed throughout. Additionally, the figures are blurry and I was unable to read them. Overall, here is redundancy in the content. Try to streamline and eliminate duplicating findings and key messages throughout.
Introduction
The introduction sets up the rest of the article- nicely done. My one comment is:
Introduction paragraph starting at line 41 is not well flushed out. These points mimic many that were in the previous paragraph. I would suggest combining these and not ending a paragraph with a question—I was expecting an answer or evidence that responds to this question and never clearly received it.
Section 2
Add description of the key policy elements: Community design n=13; movement network n=15; lot layout n=7; and public parkland n=8. Community design and movement network are not clear to the reader?
Section 3
Shorten title for this section: “Understanding the ‘policy world’ for Liveable Neighbourhoods”
Paragraph 136-Line 143 needs clarity: I do not fully understand what the Greenfield and Infill development applications are or what they have to do with LN? : It is not clear what the second “policy window” was? What are the research translation efforts—translating RESIDE to another version of LN? Perhaps the lack of clarity is because I Figure 1 is too blurry to read?
Section 4:
Interesting results!
Could you provide detail on how you from learned policy makers (e.g. learned that they wanted to know what aspects of LN were being implemented/working)? Was an advisory board formed? Were there regular meetings? This information comes in Section 6 but I wish it was here.
In the introduction of the LN, it states that the desired policy outcomes including walking, cycling and public transport use, sense of community, safety from crime and mental health. You do not present anything on cycling or public transport, only walking? Also, I would present results in the same order listed in introduction for continuity.
Section 5:
Title too vague—what are implications? This is only time that word is used and everywhere else policy-specific instead specific policy is used.
Was there any understanding of why certain aspects were implemented over others? Do we have an understanding of barrier/facilitators to certain changes in the community besides the leaky pipeline? Perhaps examples of how/why some things were implemented over others? I don’t think that you answer Brownson’s first key domain as he defines it: process- approaches to enhance the likelihood of policy adoption? Maybe you should move this to your introduction—as this frames the entire presentation of your results? In other words, doesn’t the understanding of the policy world or links help with policy adoption?
Section 6 has a lot of redundancy. The whole last paragraph can be deleted or integrated within other paragraphs.
The conclusion is great! This is important work—thank you!
Author Response
Reviewer 2
Comments and Suggestions for Authors
Overall Comments:
· This paper provides methods, lessons learned and results from a longitudinal natural experiment that rigorously evaluated the implementation process to quantify the dose of the policy and understand the relationship to the intended active living and health outcomes. The insights from this study are applicable and important for bridging the research- policy gap.
· There are several grammatical edits, typos, etc. that need to be fixed throughout. Additionally, the figures are blurry and I was unable to read them. Overall, here is redundancy in the content. Try to streamline and eliminate duplicating findings and key messages throughout.
Ø The manuscript has been proof read and checked for typos and grammatical errors throughout.
Ø We have re-produced the figures which should now be of a higher quality.
Introduction
· The introduction sets up the rest of the article- nicely done. My one comment is:
· Introduction paragraph starting at line 41 is not well flushed out. These points mimic many that were in the previous paragraph. I would suggest combining these and not ending a paragraph with a question—I was expecting an answer or evidence that responds to this question and never clearly received it.
Ø We have reviewed the paragraph, edited it and combined it with the preceding paragraph as suggested by the reviewer.
Section 2
· Add description of the key policy elements: Community design n=13; movement network n=15; lot layout n=7; and public parkland n=8. Community design and movement network are not clear to the reader?
Ø We have added a description of the different policy elements – see lines 272 – 284 which we trust now provides sufficient context for the reader to better understand the focus and content of the policy.
Section 3
· Shorten title for this section: “Understanding the ‘policy world’ for Liveable Neighbourhoods”
Ø This has been shortened and edited as suggested by the reviewer
· Paragraph 136-Line 143 needs clarity: I do not fully understand what the Greenfield and Infill development applications are or what they have to do with LN? :
Ø We have edited section 2 to better clarify and add more detail on the RESIDE study and its methods and provided a definition of “greenfield” and infill developments: “greenfield sites (i.e. previously unused or undeveloped land areas that had been rezoned, typically from urban deferred or rural to urban land uses and projects). Others were being constructed in brownfield areas (existing urban zones being redeveloped, sometimes following rezoning from industrial or other non-residential use)”.
Ø These are the types of development are that the LN policy was produced to influence the built form outcomes of.
· It is not clear what the second “policy window” was? What are the research translation efforts—translating RESIDE to another version of LN? Perhaps the lack of clarity is because I Figure 1 is too blurry to read?
Ø We had added more detail to section 3 to explain and clarify the Multiple Streams theory of policy development that we have aligned and compared the Liveable neighbourhoods policy development to. This helps to explain what we meant by the “second window of opportunity”
Section 4:
· Interesting results!
· Could you provide detail on how you from learned policy makers (e.g. learned that they wanted to know what aspects of LN were being implemented/working)? Was an advisory board formed? Were there regular meetings? This information comes in Section 6 but I wish it was here.
Ø We have moved this detail, plus more as requested by the reviewer to section 4.
Ø We have added a new section identifying how the partnership was formed and how its ongoing collaboration functioned utilised by the researchers.
· In the introduction of the LN, it states that the desired policy outcomes including walking, cycling and public transport use, sense of community, safety from crime and mental health. You do not present anything on cycling or public transport, only walking? Also, I would present results in the same order listed in introduction for continuity.
Ø The analysis of health outcomes with the detail process evaluation measures has only been conducted on the four documented outcomes – walking, sense of community, mental health and victimisation from crime (“The results revealed a strong dose–response relationship between policy compliance and four health-related outcomes”).
Ø Other RESIDE studies from the wider REISDE project did focus on outcomes such as cycling and public transport use but these were not subject to the analysis undertaken as part of the process evaluation. We have however, mentioned these in this section to highlight the broader findings with health outcomes from the RESIDE study.
Section 5:
· Title too vague—what are implications? This is only time that word is used and everywhere else policy-specific instead specific policy is used.
Ø The titles has been shortened and changed. It now reads: A case study of a natural experiment bridging the ‘research into policy’ and ‘evidence-based policy’ gap for active living science
· Was there any understanding of why certain aspects were implemented over others? Do we have an understanding of barrier/facilitators to certain changes in the community besides the leaky pipeline? Perhaps examples of how/why some things were implemented over others? I don’t think that you answer Brownson’s first key domain as he defines it: process- approaches to enhance the likelihood of policy adoption? Maybe you should move this to your introduction—as this frames the entire presentation of your results? In other words, doesn’t the understanding of the policy world or links help with policy adoption?
Ø Under section 7 we have added new detail and a discussion outlining the barriers and facilitators to policy implementation.
· Section 6 has a lot of redundancy. The whole last paragraph can be deleted or integrated within other paragraphs.
Ø We have reviewed and revised this section ensuring we have removed duplication across sections.
· The conclusion is great! This is important work—thank you!
Reviewer 3 Report
see attachment file

Author Response
Reviewer 3
· Manuscript summary and overall comments:
· The creation of healthy, active communities is a priority for environmental sustainability and health perspectives. The design of built environment has an influential impact on residents’ living behaviors. Bridging the research-translation gap between researchers and policy-makers is an important component in creating healthy built environments. The aim of this study was to provide an overview and discussion of policy-specific methodologies undertaken to quantitatively assess the implementation of the successes and impact of the Liveable Neighbourhoods policy. There are parts of this brief report where clarity is needed.
· The overall connection to public health is not clear considering this is a public health journal.
Ø We have edited the introduction, and opening paragraph, to establish the link and set the scene, between the design of the built environment and health outcomes. We also make reference in the introduction to the literature base of studies supporting a link between the design of the built environment and health behaviours and outcomes.
Ø The purpose of the commentary paper was to highlight the fact that despite all this evidence, its translation into real world policy and practice still remains rare. Our paper provides a commentary on the RESIDE project that aimed to, and was successful, in translating its research efforts and findings into a real world policy shares this experience and lessons learned.
· Critiques pertaining to this paper are described below.
Specific comments
· The author mentions that over the past 15 years, there has been a comprehensive body of research which address the impact of the design of built environments on residents’ active living behaviors. There should be some sort of delineation of the work that has been done in this field. This paper will benefit from including more studies that have been done demonstrating the impact of built environments on residents’ living behaviors.
Ø We realised we did not provide references to support this statement. As such we have added in references to a number of reviews that have summarised the extensive literature in the field (thus supporting out statement).
Ø We have also revised the introduction section to address the concerns of the reviewer – please see the revised text on line 65 onwards. We have also referenced a number of case studies and research papers which compare areas developed under different design principles or alternate planning movements or theories such as New Urbanism or Smart Growth projects and compare the health outcomes of residents.
· On Page 3, it is mentioned that in 2003, the RESIDential Environments project (RESIDE) began tracking 1803 people who were moving into one of the 74 new developments in Perth, Western Australia. It is unclear why these 1803 people were selected. There is no data on participants shown. What have we learned from these 1800 participants?
Ø We have edited section 2 to better clarify and add more detail on the RESIDE study and its methods and outlines how the 1803 participants were identified and selected.
Ø We have also provided a description of the sample demographics at baseline.
Ø In section 6 we had outlined the outcome evaluation results of the associations of policy compliance with walking, mental health, sense of community and victimisation from crime.
Ø We have also added new text in describing some further analyses that unpacked which of the policy requirement were most important for the health outcomes (line 331 onwards).
· While evaluating the implementation of the policy, it is unclear why this would be the task of health researchers. It should be city planning that determines whether the intended policy was implemented or not. As it stands, I do not see the intersection of public health in this specific aim or in this case report more generally.
Ø The reviewer is correct in that in an ideal world the planning policy would be evaluated by the department responsible for it. However, this does not often regularly (if at all) happen.
Ø Further, as outlined in the introduction, Government policy and planning initiatives determine the way cities, towns and neighbourhoods are developed and configured which in turn plays a vital role in creating and shaping the environments that support or undermine residents’ health and their ability to be safely and conveniently physically active. Moreover, the active living field has identified the evaluation of planning policies as an important focus of research, and this has been identified by the policy makers as important for research translation and application of the empirical findings.
Ø Hence, the introduction of the LN policy in Western Australia provided a unique opportunity to evaluate the impacts of a new planning policy on the housing developments being built and the health outcomes of the residents residing and mowing into those. The premise and focus of this paper is in describing the benefits and lessons learned of undertaking such a policy evaluation and its links to public health.
· Five key strategies are described in the introduction related to RESIDE. There should be an explanation for the reason why using these specific strategies are important.
Ø We have revised the paper to clarify this and in response to similar comments by the other reviewers.
Ø We have re-framed the paper bringing Brownson et al’s three policy relevant research strategies up front as a framework around which to discuss our findings. Further, we have matched Brownson’s three strategies to the five strategies identified by Giles-Corti et al as essential for closing the active living research-translation gap.
Ø These were chosen because they represent a body of work by leading active living researchers (and one of the co-authors of this paper) of potential strategies that are important for research translation and we felt it was a good framework around which to present and describe our findings and lessons learned.
· Why does this report only include evidence from 36 of the 74 housing developments? How might this influence results?
Ø The process evaluation was undertaken on a sub-set (n=36) of the 74 housing developments that the RESIDE study was focussed on. We have added some information outlining and clarifying this in section 5.
· The figures presented are not discernable (blurry) and impedes fully understanding the data. 2
Ø We have re-produced the figures which should now be of a higher quality.
· On page 7, Oliver et al. 2014 appears to be in a different format than the other references.
Ø This has been corrected.
· “Leaky pipe”: While the description of the policy pipeline process is important and highlights why bridging the research-policy gap between policy makers and researchers should be prioritized, it is should be placed in the introduction and its relationship to public health should be made clear. How does knowing where the leaks are in the pipe change the way health researchers ask questions and design their studies?
Ø We have introduced the concept of the leaky pipe in the final section of the paper as this is the culmination of our learning and experience of undertaking the policy relevant research and natural experiment and presents a framework for future studies to consider designing their methodologies around.
Ø To this end, and for this reason, we have left the introduction and discussion of the leaky pipe where it was in the paper.
Reviewer 4 Report
This paper provides a case study overview and discussion of the key lessons from the RESIDE project, mainly focusing on how to translate active living science into urban design policy and practice. The topic is very interesting. However, this paper needs to better articulate the details of the research method. Several key methodological and logical issues are not addressed in the paper. Especially, the measures of the study were not clearly described, making the rigor of this study questionable. Specific comments to strengthen this paper are as follows.
The text in Figure 1 and 2 is not legible
“In 2003, with the support of key local policymakers and advocates the RESIDential Environments project (RESIDE) began tracking 1803 people who were moving into one of 74 new housing developments in Perth, Western Australia [20], and followed them for a decade.” (P 3 Line 103-105)
Only 2003 data? Authors need to further elaborate longitudinal data.
In discussing RESIDE we will reflect on five key strategies [3] identified as essential to close the research-translation gap:: (P3 Line 111-114)
How these five gaps came from? I cannot follow the logics of why authors want to highlight these five gaps and which methodology and analyses were used. Authors need to explain the importance of these items in the introduction. Also very confused why authors only have four sub-headings.. order of these gaps and subheadings are inconsistent
“It focuses on the policy-specific methodology and analyses that 109 ensured RESIDE findings resonated with the local planning industry” (P3 Line 108-9)
Please specify the policy-specific methodology and analyses as I am unable to see any methodology and analyses in this commentary.
“Despite incomplete implementation of the policy, analyses were was also undertaken to 183 determine” (p5 line 182)
Take out “was”
4. Establish links and joint research agendas with policymakers and practitioners (p 5 line 182-192)
In this section, I was unable to see why and how authors highlight joint research agendas with policymakers and practitioners. At least, authors need to explain how the process evaluation
“Brownson and colleagues [10] suggest that policy-relevant evidence should include both quantitative and qualitative information and describe three key domains of evidence-based policy: (1) process - to understand approaches to enhance the likelihood of policy adoption; (2) content - to identify specific policy elements that are likely to be effective; and (3) outcomes - to document the potential impact of policy. All three types of information can inform the decision making of policy makers in terms of the success (or otherwise) of the policy in relation to the health and well-being outcomes” (P5 Line 199-203)
How are these three key domains of evidence-based policy from Brownson and colleagues related to RESIDE? I expected that authors attempted to elaborate the three key domains of the evidence-based policy of RESIDE..
Author Response
Reviewer 4
Comments and Suggestions for Authors
· This paper provides a case study overview and discussion of the key lessons from the RESIDE project, mainly focusing on how to translate active living science into urban design policy and practice. The topic is very interesting.
Ø We thank the reviewer for their positive appraisal of the paper.
· However, this paper needs to better articulate the details of the research method. Several key methodological and logical issues are not addressed in the paper. Especially, the measures of the study were not clearly described, making the rigor of this study questionable. Specific comments to strengthen this paper are as follows.
Ø As per a concern of reviewer #2, we have edited section 2 to better clarify and add more detail on the RESIDE study and its methods.
Ø The process evaluation measures are outlined (and referenced) in detail elsewhere. Adding full detail of the measures was beyond the scope and focus of this paper and would have added to its length.
Ø We have added a synthesis of the four design elements in section 5 which provides an overview of the types of design features that were measured under each of the four policy elements, plus examples of the types of design features measured.
· The text in Figure 1 and 2 is not legible
Ø We have re-produced the figures which should now be of a higher quality.
· “In 2003, with the support of key local policymakers and advocates the RESIDential Environments project (RESIDE) began tracking 1803 people who were moving into one of 74 new housing developments in Perth, Western Australia [20], and followed them for a decade.” (P 3 Line 103-105)
· Only 2003 data? Authors need to further elaborate longitudinal data.
Ø We have added detail on the methods of the RESIDE project, including that of the participants recruited and their follow up, as respective sample sizes at each time point (lines 135-146).
· In discussing RESIDE we will reflect on five key strategies [3] identified as essential to close the research-translation gap:: (P3 Line 111-114) How these five gaps came from? I cannot follow the logics of why authors want to highlight these five gaps and which methodology and analyses were used. Authors need to explain the importance of these items in the introduction. Also very confused why authors only have four sub-headings.. order of these gaps and subheadings are inconsistent
Ø We have sorted the issue with the numbering of the sections and sub-headings to make this consistent.
Ø We have edited the section to better explain out rationale for the use of the five strategies identified by Giles-Corti et al as essential for closing the active living research-translation gap.
Ø These were chosen because they represent a body of work by leading active living researchers (and one of the co-authors of this paper) of potential strategies that are important for research translation and we felt it was a good framework around which to present and describe our findings and lessons learned.
· “It focuses on the policy-specific methodology and analyses that 109 ensured RESIDE findings resonated with the local planning industry” (P3 Line 108-9)
· Please specify the policy-specific methodology and analyses as I am unable to see any methodology and analyses in this commentary.
Ø We have made substantial edits to section 2 to better explain the methodology and process evaluation approach undertaken. We have presented this as a brief overview to fit with the focus of the commentary, citing the relevant references that present the methods in full detail.
· “Despite incomplete implementation of the policy, analyses were was also undertaken to 183 determine” (p5 line 182). Take out “was”
Ø Thank you – this was a typo and has been edited / deleted.
· 4. Establish links and joint research agendas with policymakers and practitioners (p 5 line 182-192)
· In this section, I was unable to see why and how authors highlight joint research agendas with policymakers and practitioners. At least, authors need to explain how the process evaluation
Ø We have made substantial edits to section 4 to better explain how we established the joint research agendas with our partner policy makers.
Ø We have also expanded on our explanation of the methods of the process evaluation (section 2) and the benefits of these data and outputs (section 8 and 9).
· “Brownson and colleagues [10] suggest that policy-relevant evidence should include both quantitative and qualitative information and describe three key domains of evidence-based policy: (1) process - to understand approaches to enhance the likelihood of policy adoption; (2) content - to identify specific policy elements that are likely to be effective; and (3) outcomes - to document the potential impact of policy. All three types of information can inform the decision making of policy makers in terms of the success (or otherwise) of the policy in relation to the health and well-being outcomes” (P5 Line 199-203). How are these three key domains of evidence-based policy from Brownson and colleagues related to RESIDE? I expected that authors attempted to elaborate the three key domains of the evidence-based policy of RESIDE..
Ø We have revised the paper to clarify this and in response to similar comments by the other reviewers.
Ø We have re-framed the paper bringing Brownson et al’s three policy relevant research strategies up front as a framework around which to discuss our findings. Further, we have matched Brownson’s three strategies to the five strategies identified by Giles-Corti et al as essential for closing the active living research-translation gap.
Round 2
Reviewer 3 Report
The revisions are fine.
Author Response
We thank the reviewer for their consideration of the manuscript.
We note they have indicated "the revisions are fine", but that an "English language and style are fine/minor spell check" is required.
We have employed the services of a professional editor who had reviewed the manuscript and edited it thoroughly for language style and spelling.
We trust this issue should now be resolved.
Reviewer 4 Report
Thank you for addressing the comments!
Author Response
We thank the reviewer for their consideration of the manuscript.
We note they have not indicated and specific further revisions that are required in their comments made, but have indicated that an "English language and style are fine/minor spell check" is required.
We have employed the services of a professional editor who had reviewed the manuscript and edited it thoroughly for language style and spelling.
We trust this issue should now be resolved.